# Pooling saliva samples as an excellent option to increase the surveillance for SARS-CoV-2 when re-opening community settings

Joaquín Moreno-Contreras[1], Marco A. Espinoza[1], Carlos Sandoval-Jaime[1], Marco A. Cantú-Cuevas[2], Daniel A. Madrid-González[2], Héctor Barón-Olivares[3], Oscar D. Ortiz-Orozco[3], Asunción V. Muñoz-Rangel[3], Cecilia Guzmán-Rodríguez[3], Manuel Hernández-de la Cruz[3], César M. Eroza-Osorio[3], Carlos F. Arias[1], Susana López[1]*

1 Departamento de Genética del Desarrollo y Fisiología Molecular, Instituto de Biotecnología UNAM, Cuernavaca, Morelos, México, 2 Secretaría de Salud del Edo. de Morelos, Cuernavaca, Morelos, México, 3 Servicios de Salud del Edo. de Morelos, Cuernavaca, Morelos, México

* susana@ibt.unam.mx

## Abstract

In many countries a second wave of infections caused by the severe acute respiratory syndrome coronavirus 2 (SARS-CoV-2) has occurred, triggering a shortage of reagents needed for diagnosis and compromising the capacity of laboratory testing. There is an urgent need to develop methods to accelerate the diagnostic procedures. Pooling samples represents a strategy to overcome the shortage of reagents, since several samples can be tested using one reaction, significantly increasing the number and speed with which tests can be carried out. We have reported the feasibility to use a direct lysis procedure of saliva as source for RNA to SARS-CoV-2 genome detection by reverse transcription quantitative-PCR (RT-qPCR). Here, we show that the direct lysis of saliva pools, of either five or ten samples, does not compromise the detection of viral RNA. In addition, it is a sensitive, fast, and inexpensive method that can be used for massive screening, especially considering the proximity of the reincorporation of activities in universities, offices, and schools.

## Introduction

After more than one year of the COVID-19 global health emergency, the early detection of severe acute respiratory syndrome coronavirus 2 (SARS-CoV-2) remains a key factor to decrease community virus spreading. Although several antigenic and immunologic assays have been developed, the amplification of specific regions of the viral genome by reverse transcription quantitative-PCR (RT-qPCR) in nasopharyngeal swabs (NPS) remains the golden standard for SARS-CoV-2 diagnosis [1–3]. However, due to the pandemic there has been a shortage of reagents used for testing, including swabs, viral transport medium, and kits for viral RNA extraction, limiting test capabilities in many countries with an active viral propagation.

Recently, we demonstrated that a direct lysis procedure to prepare RNA from saliva samples is a feasible method to detect the SARS-CoV-2 genome, and as efficient as column-based

**Data Availability Statement:** All relevant data are within the paper.

**Funding:** Part of the reagents used in this study were provided by the Instituto Nacional de Diagnóstico y Referencia Epidemiológica, supported by INSABI. This work was supported by grant 314343 from CONACyT to SL. JMC was a recipient of a scholarship from CONACyT. The funders had no role in study design, data collection and analysis, decision to publish, or preparation of the manuscript.

**Competing interests:** The authors have declared that no competing interests exist.

methods, with a significant reduction in costs and time of sample processing [4]. Saliva is a clinical specimen that has been approved for emergency use by the Food and Drug Administration (FDA) for SARS-CoV-2 diagnosis; since it can be self-collected, there is a reduced risk of healthcare workers involved in sampling, making it a good candidate to increase the amount of tests performed in regions with shortages of personal protection equipment (PPE) supplies [5,6]. Pooling of samples has been implemented as a diagnostic tool for other viruses; if a pool is negative, all samples are considered to be below the limit of detection of the test, whereas when a pool is positive, the samples are evaluated individually. This strategy allows to test large number of samples more efficiently and with a reduced cost. For SARS-CoV-2 detection, pooling of samples has been evaluated using NPS and oropharyngeal swabs (OPS), as well as saliva samples, allowing to save reagents, increasing the amount of tests performed and reducing costs, especially in regions with a low prevalence of the virus [7–10]. Even though pooling offers some advantages, sensitivity can be compromised by several factors, including pool size, amount of sample analyzed, and RNA extraction. In this study, we evaluated 1,086 saliva specimens of ambulatory patients in pools of five or ten samples by RT-qPCR. Initially, positive individual samples with a known $C_T$ value were mixed with either 4 or 9 negative samples, and the RNA in the pools was obtained by a lysis protocol as previously reported [4], and used directly for the RT-qPCR test. The $C_T$ value obtained for each pool was compared with that of the positive sample used in the pool. We found that the sensitivity decreased in pools of ten samples, while in pools of five samples the sensitivity was not significantly affected. We propose that saliva pooling and its direct lysis is a good method to detect SARS-CoV-2 that will help to increase the amount of tests performed and accelerate diagnosis at a reduced cost, particularly now, that several public spaces and schools are reopening.

## Materials and methods

### Sample collection

1,086 saliva samples were collected from August 7[th] to October 30[th] 2020 by healthcare workers from the Epidemiology Department of the Health Ministry of the State of Morelos (Secretaría de Salud Morelos, SSM). All samples were taken from ambulatory patients as part of the government program "Pruebas COVID-19 en tu comunidad", aiming to bring SARS-CoV-2 tests into communities located far apart from Cuernavaca, the capital city.

### Saliva collection

Saliva was self-collected as previously described [4]. Briefly, patients were asked to spit 2–3 ml of saliva into sterile urine cup containers containing 1 ml of viral transport medium (MTV). After collection, samples were stored and kept at 4˚C until transported to the Instituto de Biotecnología/UNAM (IBT/UNAM) for their analysis, within the next 24–48 h after sample collection.

### Saliva pooling, RNA extraction and RT-qPCR

Five or ten saliva samples were pooled by mixing 50 µl of each sample reaching a final volume of 250 or 500 µl, respectively. Pools were mixed homogeneously by pipetting several times and kept at room temperature (RT) until processing. Saliva pools or individual samples were either treated with Quick Extract[TM] DNA Extraction Solution (QE; Lucigen) by mixing 50 µl of saliva samples with 50 µl of the QE reagent, as reported [4], or with 15 µl of Proteinase K (20 mg/ml, Invitrogen) when using the SalivaDirect protocol [11]. The mixtures were heated for 5 min at 95˚C; and then cooled on ice and kept at 4˚C until use (within 1 h of QE, or proteinase K treatment).

SARS-CoV-2 detection was performed using the Berlin protocol, using the reported oligo-nucleotides and probes for viral gene E and for human RNase P [12]. The RT-qPCRs were performed using the StartQ one-step RT-qPCR (Genes2 life) kit, using 2.5 μl of the QE- treated saliva in 22.5 μl of RT-qPCR reaction mixture, or 5 μl of proteinase K-treated saliva in 20 μl of RT-qPCR reaction mixture. Samples were analyzed in an ABI 7500 sequence detector system (Applied Biosystem) with the following thermal protocol: 50˚C for 15 min, 95˚C for 2 min and then 45 cycles of 95˚C for 15 s and 60˚C for 30 s. Individual and pools of five samples with a threshold cycle ($C_T$) equal to or less than 38 were classified as positive. Pools of ten samples with a $C_T$ equal to or less than 41 were classified as positive. The change of $C_T$ between pooled and unpooled samples was calculated by subtracting the $C_T$ of viral gene E in the pool from the $C_T$ of the individual unpooled sample ($C_{T\ change} = C_{T\ pool} - C_{T\ unpooled}$). In pools with more than one positive sample, the highest $C_T$ value of the individual unpooled sample was taken.

## Statistical analysis

Statistical analysis was performed using GraphPad Prism 6.0 (GraphPad Software Inc.) as described in Results.

## Ethical considerations

The protocol used in this study was conducted under the ethical principles and approval of the Bioethics Committee of the Instituto de Biotecnología (Project # 393) of the National University of Mexico (UNAM). Verbal informed consent was obtained from all individuals enrolled in this study and was witnessed by personnel of the Health Ministry of the State of Morelos, who were in charge of collecting the samples.

## Results

### Effect of saliva sample pooling on the sensitivity of the assay

To evaluate the effect of pooling samples on the $C_T$ value for detection of SARS-CoV-2, positive saliva samples with different $C_T$ values (ranging from 24.2 to 37) for viral gene E were mixed either with four, or nine virus-negative saliva samples. Equal amounts of each sample were homogenously mixed to prepare the pools, and the RNA was directly obtained from a 50 μl aliquot of the pooled samples using the QE lysis buffer (Lucigen) and boiling for 5 min, as reported [13].

A slight decrease in the $C_T$ value of the positive saliva samples was observed when it was determined in the context of the pools of five samples, with a mean change in $C_T$ of 1.7 units (95% C.I: 0.8, 2.6, lineal regression, $R^2$:0.9388, p<0.0001). In the pools of ten samples the $C_T$ value decreased in average 2.6 units (95% C.I: 1.7, 3.5, lineal regression, $R^2 = 0.9214$, p<0.0001). In the five-sample pools, 100% (10/10) of the positive samples were detected, while in the pools of ten samples, a sample with a $C_T = 37$ was not detected (Fig 1A). As a control, negative saliva samples were tested in pools of five or ten.

### Evaluation of saliva sample pools from ambulatory patients

To evaluate saliva pooling and its direct lysis as a diagnostic tool, the presence of SARS-CoV-2 genome was determined in 1,075 saliva specimens from patients having two or more symptoms related to COVID-19 [14,15]. Samples were collected from ambulatory patients on eleven mobile medical units in 89 locations that belong to the Jurisdicción Sanitaria N.2 in Morelos, México. The presence or absence of SARS-CoV-2 genome was detected by RT-qPCR, as

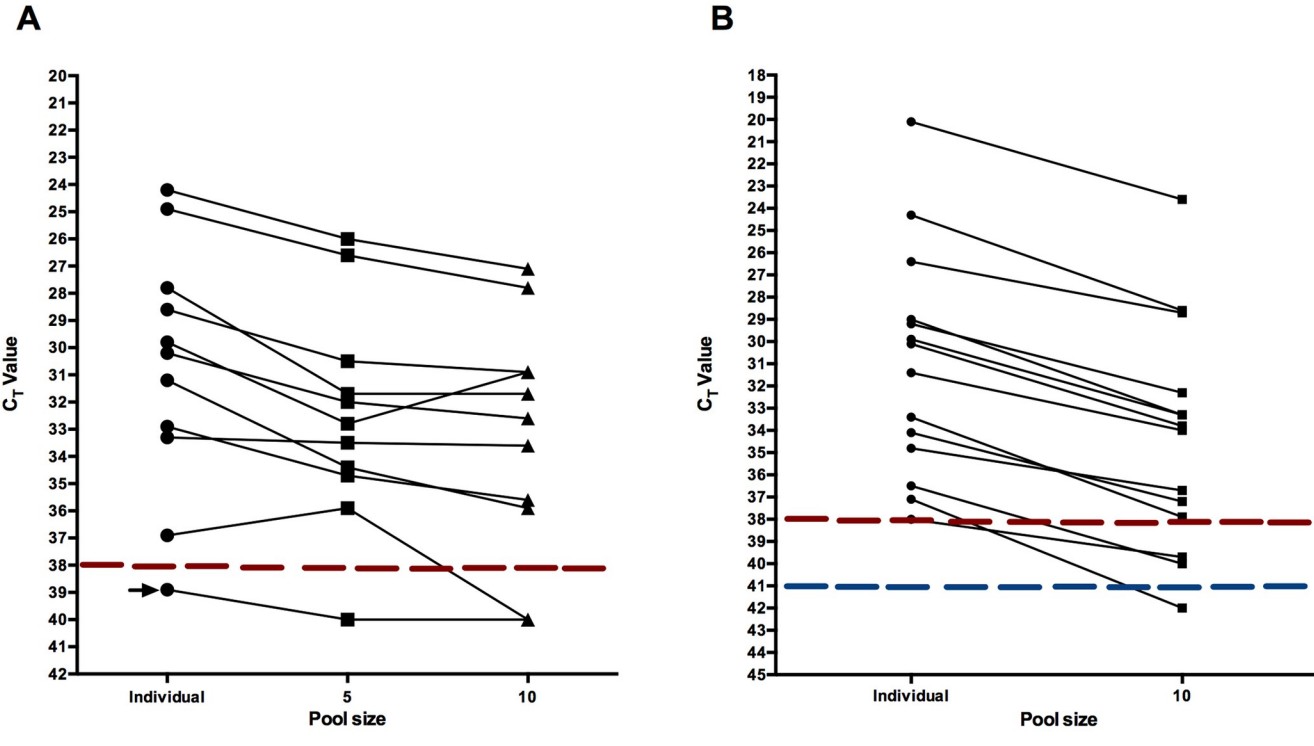

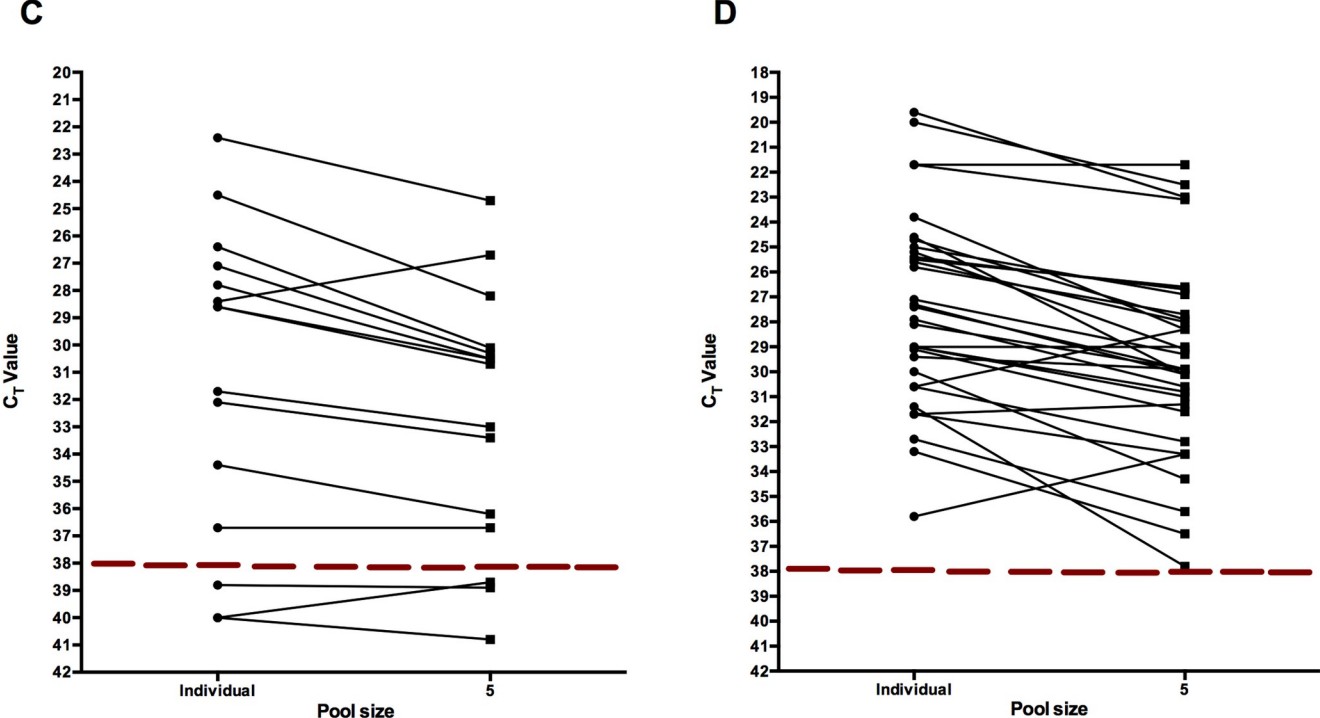

**Fig 1. Detection of SARS-CoV-2 in pools of five and ten saliva samples.** A) SARS-CoV-2 positive saliva samples were mixed with either four or nine negative samples. The $C_T$ value of the individual samples was compared to that obtained in the pooled samples; the mean change of $C_T$ was +1.7 (95% C.I:

0.8, 2.6, lineal regression, $R^2$:0.9388, p<0.0001), and +2.6 (95% C.I: 1.7, 3.5, lineal regression, $R^2$ = 0.9214, p<0.0001) units for pools of 5 and 10, respectively. RNA was obtained from pools of ten (B) or five (C) previously undiagnosed patient saliva samples using QE lysis buffer; positive pools were analyzed as individual samples ($C_T$ mean change for pool of ten, +3.2 $C_T$, 95% C.I: 2.7, 3.9; pool of five +1.8 $C_T$, 95% C.I: 0.5, 2.3). D) As an alternative method of RNA extraction, pools of five undiagnosed saliva samples were treated with proteinase K, and positive pools were analyzed individually ($C_T$ mean change + 2.2 $C_T$, 95% C.I: 1.4, 2.7). In all figures, the $C_T$ value of the viral gene E obtained in individual, or pools of five or ten saliva specimens are represented by lines connecting each condition. Dotted red lines represent $C_T$ cut-off value = 38; for pools of ten samples this value is represented by blue dotted lines, with a $C_T$ cut-off value = 41. Negative control in panel A is shown with an arrow.

described in the Materials and Methods section. Positive pools were deconvoluted and analyzed as individual samples.

From the total saliva specimens collected, 260 were analyzed in pools of ten; 12 of the 26 pools resulted negative, having a $C_T$ value equal or higher than 41. Since we had previously observed that pooling 10 samples decreased the $C_T$ value by approx. 2.7 units, pools with a $C_T$ < = 41, in which a smooth sigmoidal amplification curve was additionally obtained, were taken as positive. Amplification of the viral gene E was detected in the remaining 14 pools, and individual saliva samples were then tested from these pools. Seven pools contained one positive sample, two pools contained two positive samples, four pools contained 3 positive samples, and 1 pool contained 4 positive samples. Comparing the $C_T$ value of the pools with that obtained with individual samples, the mean change of $C_T$ was +3.2 (95% C.I: 2.7, 3.9, Fig 1B).

Subsequently, 235 saliva samples were analyzed in pools of 5; 32 of the 47 pools analyzed were negative. Of the 15 positive pools, 10 had one positive sample, and 5 contained 2 positive samples. When the $C_T$ of the individual samples was compared to that obtained in the pooled samples, the mean difference of $C_T$ was +1.8 units (95% C.I: 0.5, 2.3, Fig 1C).

To test an alternative method of RNA extraction that has been recently described for this purpose [11], 580 saliva specimens grouped in 116 pools of 5 samples were treated with proteinase K and boiled for 5 min, as described [11]. In this assay, we found 84-negative, pools, and the remaining 32 pools were positive. When these pools were analyzed individually, 23 pools contained one positive sample, 6 had 2 positive samples and 3 contained 3 positive samples. Comparing the $C_T$ values of the pools with those obtained with individual samples, the mean change was +2.2 $C_T$ units (95% C.I: 1.4, 2.7, Fig 1D). When the change in $C_T$ values obtained with the QE buffer or proteinase K treatments were compared, no significant differences were found. Accordingly, when the samples in pools of 5 with a $C_T$ value between 38.8 and 41 were analyzed individually, a $C_T$ >38 was found in the samples (Fig 1C).

Using the strategy of saliva pooling, samples with a $C_T$ value close to the cut-off (>35) could be lost; however, an analysis of the distribution of the $C_T$ values obtained from 436 positive samples detected in our laboratory, showed that less than 8.5% of the samples analyzed had a $C_T$ > 35 (6% had a $C_T$ = 37, and 2.5% had a $C_T$ = 38), while the majority of the samples analyzed (66.9%) had $C_T$ values between 26 and 35 (Fig 2).

As part of the re-opening activities of our Institute (Instituto de Biotecnología), asymptomatic students and workers were tested for SARS-COV-2 in a pilot study from the 4[th] to the 15[th] of January, prior to their incorporation to work, using pooled saliva specimens. For this, 910 saliva samples were analyzed in 182 pools of five samples each and RNA was obtained by QE-direct lysis. We detected 177-negative, and 5-positive pools, allowing the detection of 6 positive samples (representing a positivity of 0.6%). This enabled the isolation the of positive individuals, preventing the spread of the virus in our community. Additionally, using this protocol 77.9% of reactions were saved.

## Discussion

Vaccines against SARS-CoV-2 are a key factor to control viral transmission, but even though several programs of vaccination are being implemented around the world, their cost,

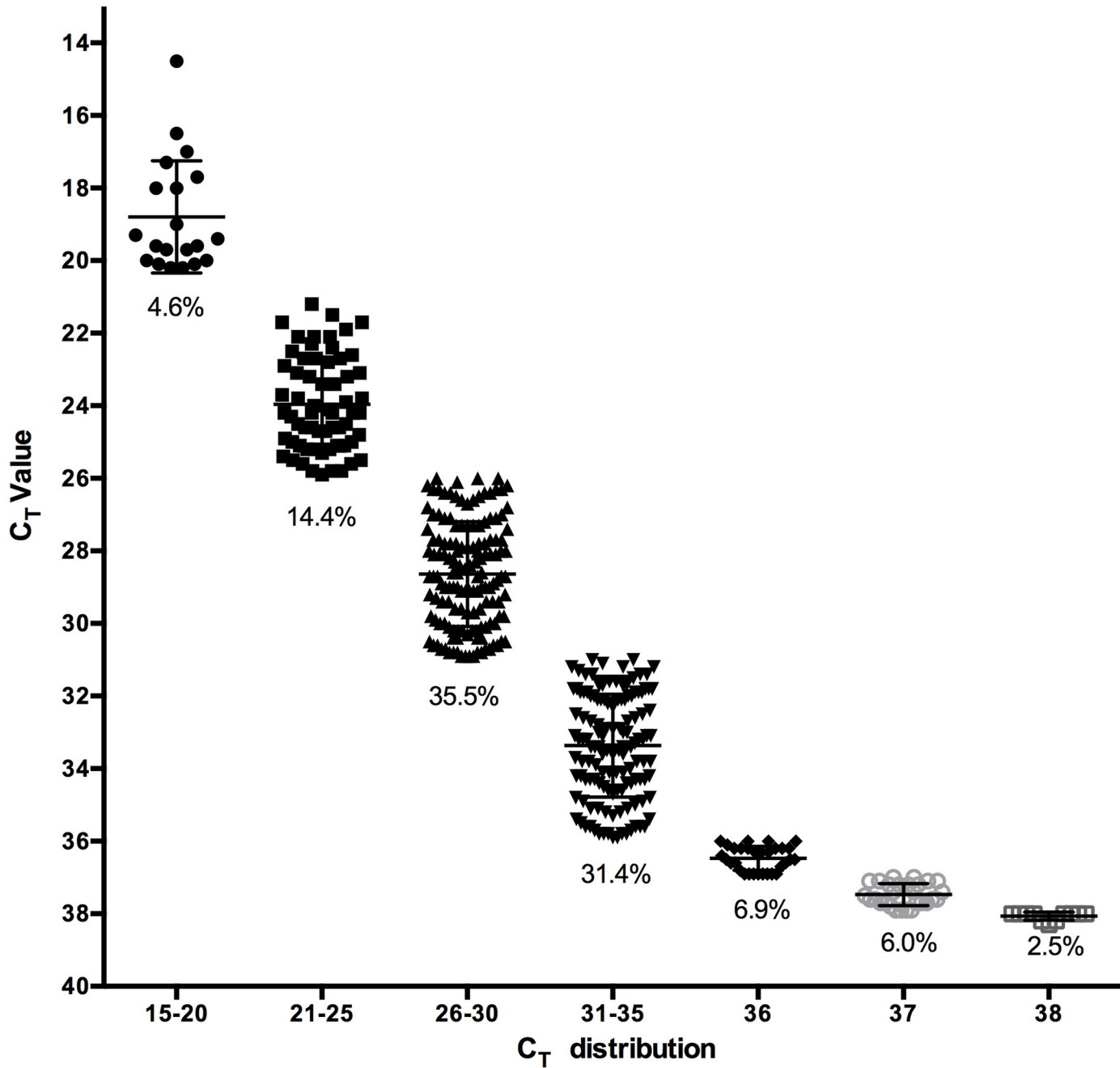

**Fig 2. Distribution of $C_T$ values in positive samples.** $C_T$ value of viral gene E from 436 positive samples are represented in intervals of five $C_T$s, with exception of $C_T$ = 36, 37 and 38. The percentage from the total number of samples analyzed is indicated in the figure.

availability, and distribution are a bottleneck, especially for developing countries. As long as susceptible populations are not covered by vaccination, detection of infected people needs to be continued to prevent spreading of the virus. Saliva pooling represents a viable strategy to increase testing capabilities with a reduced cost, and unlike antigen tests, the specificity and sensibility are not compromised. In our study, when using pools of ten samples 36.1% of reactions were saved, while 51.1% of reactions were saved in pools of five samples (Table 1). These results were obtained from symptomatic patients with a positivity of 9.5%, however in

**Table 1. Summary of results obtained from pools of five and ten saliva samples from patients suspected to have COVID-19.**

| Pooled samples | | | | | | |
|---|---|---|---|---|---|---|
| Pool Size | Extraction Reagent | Samples (#) | Negative pools | Positive pools | Reactions used | Saved reactions |
| 5 | QE/Proteinase K | 815 | 116 | 47 | 398 | 417 (51.1%) |
| 10 | QE | 260 | 12 | 14 | 166 | 94(36.1%) |
| | Total | 1,075 | | | | |

QE, Quick Extract$^{TM}$ DNA Extraction Solution; #, number.

populations with a positivity < = 1% a reduction of approximately 80% in the cost of the assays is expected [16].

Saliva is a good specimen for SARS-CoV-2 detection in symptomatic and asymptomatic patients [17]. Sample pooling has been implemented to diagnose viruses like HIV, and influenza, among others [18,19]; this strategy allows to screen the prevalence of different infections in large populations, decreasing diagnostic costs and saving supplies. Detection of SARS-CoV-2 in pools has been characterized using viral RNA obtained from either NPS or OPS, or in combination; saliva samples have also been used. Different strategies for pooling have been tested, including pooling RNAs extracted from individual samples, or pooling the samples before RNA extraction [9,20,21]. In either case, column-based, commercial RNA purification kits have been used. In this work, we showed the feasibility of obtaining good quality RNA from pooled samples by a direct lysis protocol using either the QE buffer (Lucigen) or a proteinase K treatment [11], reducing time and costs of sample processing.

An important factor to consider is the number of samples to pool, which depends on the prevalence of SARS-CoV-2 in the population to study [16]. Different programs to calculate the optimal pool size have been reported [16,22], but the number of infected individuals detected in a short period of time previous to the sampling is a key factor to determine the appropriate pool size. Pools of 32, 20, 15, 10 and 5 samples have been used [20], however, pools of 5 and 10 specimens seem to affect minimally the $C_T$ value of a single positive sample in the pool; the maximum change detected in these assays was an increase of 3 $C_T$ units [10]. In this study, we found $C_T$ changes of ~2 units for five-sample pools, in accordance with previous studies [9,23]. Problems in the detection of samples with $C_T$ values higher that 35 have been reported for ten-sample pools [23,24], however, here we found that our method allowed to detect positive samples with $C_T$ values equal or higher than 35.

It is interesting to note that when a correlation between viral load (expressed as $C_T$) and infectiousness (as determined by cell-culture of the samples) has been studied, it has been found that detection of SARS-CoV-2 in cell culture decreases to 20% for samples with $C_T$ > 30, and to 3% for $C_T$ = 35, suggesting that positive patients with values of $C_T$ > 35 have a very low viral load, and most probably are not infectious [25].

When a ten-fold dilution of a positive viral control used in our assays was evaluated by RT-qPCR, an increase of approximately 3.3 $C_T$ units was observed compared to the undiluted control, as expected [26]; thus, we propose to rise the $C_T$ cut-off value (from 38 to 41) when pools of ten samples are analyzed, to increase the detection of samples with $C_T$ values = > 35.

Saliva sampling is a noninvasive method with several advantages for patients and health care workers compared with NPS and OPS, and suitable for the screening of healthy individuals [8,17]. Several studies have compared the efficiency of detection of SARS-CoV-2 in saliva versus OPS and NPS and it is clear that saliva samples contain similar levels of SARS-CoV-2 genome copies as those found in NPS, and perform better than OPS [4,27,28]. In conclusion, saliva pooling and its direct lysis of the samples offers a sensitive, fast, and inexpensive method

for massive screening in the gradual de-escalation of lockdown, especially in the reincorporation of activities in universities, offices, and schools.

## Acknowledgments

We are grateful to the healthcare workers of Servicios Estatales de Salud de Morelos for their invaluable help in collecting the samples, and to the personnel of the Laboratorio Estatal de Salud Pública del Estado de Morelos, for their support in the preparation and transporting of the samples. The work of P. Gaytán, E. López and J. Yañez from the DNA sequencing and synthesis unit is also acknowledged.

## Author Contributions

**Conceptualization:** Joaquín Moreno-Contreras, Carlos F. Arias, Susana López.

**Formal analysis:** Carlos F. Arias.

**Investigation:** Joaquín Moreno-Contreras, Marco A. Espinoza, Carlos F. Arias.

**Methodology:** Joaquín Moreno-Contreras, Marco A. Espinoza, Carlos Sandoval-Jaime, Manuel Hernández-de la Cruz.

**Resources:** Carlos Sandoval-Jaime, Marco A. Cantú-Cuevas, Daniel A. Madrid-González, Héctor Barón-Olivares, Oscar D. Ortiz-Orozco, Asunción V. Muñoz-Rangel, Cecilia Guzmán-Rodríguez, Manuel Hernández-de la Cruz, César M. Eroza-Osorio.

**Supervision:** Héctor Barón-Olivares, Asunción V. Muñoz-Rangel, Cecilia Guzmán-Rodríguez, César M. Eroza-Osorio, Susana López.

**Validation:** Marco A. Cantú-Cuevas, Daniel A. Madrid-González, Susana López.

**Writing – original draft:** Joaquín Moreno-Contreras, Susana López.

**Writing – review & editing:** Carlos F. Arias, Susana López.

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
