## [Decision Letter · Decision Letter 0]

8 Dec 2021

PONE-D-21-21324

POOLING SALIVA SAMPLES AS AN EXCELLENT OPTION TO INCREASE THE SURVEILLANCE FOR SARS-COV-2 WHEN RE-OPENING COMMUNITY SETTINGS

PLOS ONE

Dear Dr. Lopez,

Thank you for submitting your manuscript to PLOS ONE. After careful consideration, we feel that it has merit but does not fully meet PLOS ONE’s publication criteria as it currently stands. Therefore, we invite you to submit a revised version of the manuscript that addresses the points raised during the review process.

We look forward to receiving your revised manuscript.

Kind regards,

Purvi Purohit

Academic Editor

PLOS ONE

 “This work was supported by grant 314343 from  Consejo Nacional de Cienciia y Tecnología (CONACyT) awarded to SL.  JMC was a recipient of a scholarship from CONACyT.” 

“We are grateful to the healthcare workers of Servicios Estatales de Salud de Morelos for their invaluable help in collecting the samples, and to the personnel of the Laboratorio Estatal de Salud Pública del Estado de Morelos, for their support in the preparation and transporting of the samples. The work of P. Gaytán, E. López and J. Yañez from the DNA sequencing and synthesis unit is also acknowledged. Part of the reagents used in this study were provided by the Instituto Nacional de Diagnóstico y Referencia Epidemiológica, supported by INSABI. This work was supported by grant 314343 from CONACyT.  JMC was a recipient of a scholarship from CONACyT.”

“This work was supported by grant 314343 from  Consejo Nacional de Cienciia y Tecnología (CONACyT) awarded to SL.  JMC was a recipient of a scholarship from CONACyT.”

Reviewers' comments:

Reviewer's Responses to Questions

**Comments to the Author**

1. Is the manuscript technically sound, and do the data support the conclusions?

Reviewer #1: Yes

2. Has the statistical analysis been performed appropriately and rigorously? 

Reviewer #1: Yes

3. Have the authors made all data underlying the findings in their manuscript fully available?

Reviewer #1: Yes

4. Is the manuscript presented in an intelligible fashion and written in standard English?

Reviewer #1: Yes

5. Review Comments to the Author

Reviewer #1: The authors covered the feasibility of pooling saliva samples for surveillance of SARS-CoV-2 in community settings. Overall, the manuscript is well-written, covers all the necessary details and data is presented appropriately. The manuscript can be accepted after addressing a couple of issues.

1. Method: RNA extraction and RT-qPCR

The authors said they have used QuickExtract DNA Extraction Solution, which is probably used to extract gDNA for direct PCR? Kindly clarify.

2. The text can benefit from a thorough proofreading since there are spelling, grammar and punctuation issues throughout the manuscript.

6. PLOS authors have the option to publish the peer review history of their article (what does this mean?). If published, this will include your full peer review and any attached files.

Reviewer #1: No

---

## [Author Response · Author response to Decision Letter 0]

13 Dec 2021

Updated Funding Statement

Part of the reagents used in this study were provided by the Instituto Nacional de Diagnóstico y Referencia Epidemiológica, supported by INSABI. This work was supported by grant 314343 from CONACyT to SL. JMC was a recipient of a scholarship from CONACyT. 

Amended Role of Funder statement

4. We note that you have included the phrase “data not shown” in your manuscript. 

The phrase was deleted from the manuscript (page 9, line 223). It was left there unintentionally, but all the information is provided in the text.

Response to Reviewer Comments 

Reviewer #1:

The authors covered the feasibility of pooling saliva samples for surveillance of SARS-CoV-2 in community settings. Overall, the manuscript is well-written, covers all the necessary details and data is presented appropriately. The manuscript can be accepted after addressing a couple of issues.

We acknowledge the supportive comments of the reviewer.

1. Method: RNA extraction and RT-qPCR

The authors said they have used QuickExtract DNA Extraction Solution, which is probably used to extract gDNA for direct PCR? Kindly clarify.

Although Quick Extract DNA Extraction Solution (QE) is a reagent employed to extract DNA, it has been shown that it is also suitable for RNA extraction, and the RNA obtained from this treatment can be directly used in RT-qPCR reactions (1). We, and others have reported the effectiveness of QE for diagnosis of SARS-CoV-2 using saliva or swabs as source of viral genome (2)(3).

References

1. Kouranova E, Forbes K, Zhao G, Warren J, Bartels A, Wu Y, et al. CRISPRs for Optimal Targeting: Delivery of CRISPR Components as DNA, RNA, and Protein into Cultured Cells and Single-Cell Embryos. Hum Gene Ther. 2016 Apr 19;27(6):464–75. 

2. Moreno-Contreras J, Espinoza MA, Sandoval-Jaime C, Cantú-Cuevas MA, Barón-Olivares H, Ortiz-Orozco OD, et al. Saliva sampling and its direct lysis, an excellent option to increase the number of SARS-CoV-2 diagnostic tests in settings with supply shortages. J Clin Microbiol. 2020;58(10):1–6. 

3. Ladha A, Joung J, Abudayyeh OO, Gootenberg JS, Zhang F. A 5-min RNA preparation method for COVID-19 detection with RT-qPCR. Medrxiv. 2020;1–3. 

2. The text can benefit from a thorough proofreading since there are spelling, grammar and punctuation issues throughout the manuscript.

We are very sorry for the mistakes in the text, we have proofread the manuscript and corrected all the spelling, grammar, and punctuation.

---

## [Editor Report · Decision Letter 1]

13 Jan 2022

POOLING SALIVA SAMPLES AS AN EXCELLENT OPTION TO INCREASE THE SURVEILLANCE FOR SARS-COV-2 WHEN RE-OPENING COMMUNITY SETTINGS

PONE-D-21-21324R1

Dear Dr. Susana Lopez

We’re pleased to inform you that your manuscript has been judged scientifically suitable for publication and will be formally accepted for publication once it meets all outstanding technical requirements.

Kind regards,

Purvi Purohit

Academic Editor

PLOS ONE

Additional Editor Comments (optional):

I congratulate the authors for drafting this manuscript so well.